# A Collaborative Framework for Customized E-Learning Services by Analytic Hierarchy Processing

**Frank S. C. Tseng** [1,*] **, Chao-Tien Yeh** [1] **and Annie Y. H. Chou** [2]

1   Department of Information Management, National Kaohsiung University of Science and Technology, 1 University Road, Kaohsiung 824, Taiwan; goden0312@gmail.com
2   Department of Computer and Information Science, ROC Military Academy, Kaohsiung 830, Taiwan; imyhchou@gmail.com
*   Correspondence: imfrank@nkust.edu.tw; Tel.: +886-7-6011-000 (ext. 34113)

**Abstract:** Thanks to the drastic proliferation of the Internet, e-learning has been recognized as an effective medium for various kinds of aggressive learners. However, due to the deficiencies of tutoring and guiding functionalities in current learning platforms, casual learners may deviate from the original course direction with frustration, when confronting inflexible course materials and fixed learning models. In the post-COVID-19 era, we believe that the most important functionality for a personal learning environment (PLE) to offer is a course recommendation process which adaptively provides a versatile course combination scheme for different learners from different perspectives. In this paper, we propose a flexible framework for users to customize their e-learning environment based on a two-stage Analytical Hierarchical Processing (AHP) structure for building adaptive course portfolios, which adaptively provides a versatile course scheme for different learners. The main objective of our framework is to transform a learner from a role of passively accepting the course content organized by instructors, into another role of proactively selecting the courses and contributing their knowledge to continuously improve the learning platform. We believe the approach proposed is a versatile way for supporting various challenges for the next generation of personal e-learning environment.

**Keywords:** AHP; course portfolio; e-learning 2.0; multi-dimensional modeling; personal learning environment (PLE)

## 1. Introduction

The rapid growth of the Internet with cloud services has accelerated the speed of knowledge creation and delivery. It also offers an efficient way for scholars and scientists to pool information and share knowledge through cyberspace. To catch up with the evolution of progressive technologies, people have to keep on learning new knowledge continuously. That inspires enterprises to establish e-learning Websites for versatile knowledge workers, and these platforms have been gradually recognized as effective and important media for lifelong learning in the past decade.

Due to the COVID-19 pandemic, more and more courses have been transferred or moved to the Internet as online courses, the e-learning service is supposed to be increasingly popular in the future, with an ever-increasing number of digitized courses cropping up everywhere in cyberspace. However, most of the traditional online learning structures provide little ways of interaction between instructors and learners, where course contents are usually unilaterally presented, and often seem to be just delivering *information* instead of delivering *learning*. Therefore, due to the deficiencies of tutoring and guiding functionalities in such learning platforms, casual learners may deviate from the original course direction, or even lose their ways on the road of advancement, when confronting inflexible learning models. We believe the most important key factor to make personal e-learning services successful is to offer a nimble mechanism for casual learners to establish their own

customized learning platforms with auxiliary guidance, such that casual learners are free from deviation of the original course direction, or learning with frustration.

As pointed out by Attwell [1], Personal Learning Environment (PLE) is becoming a promising concept driven by the emergence of ubiquitous computing technology and the development of social media software. As personal learning takes place in different contexts and situations, and is not provided by a single learning provider, PLE is not supposed to be only a software application. Instead, it was more of a new approach to using technologies for learning.

In this paper, based on the concept of the Analytical Hierarchy Processing (AHP) model [2], we intend to propose a highly flexible framework for the next generation of PLE e-learning environment, which will be interchangeably regarded as *E-learning 2.0* in the following (although such a preliminary concept has been expounded by Downes (2005) [3] and Karrer (2006) [4]).

Analytical Hierarchical Processing (AHP) is a well-known methodology invented by Saaty (1980) [2], used for making complex decisions with benefits, opportunities, costs, and risks, and for combining them to obtain an overall outcome. AHP can be combined with other models for multi-criteria decision support; e.g., fuzzy AHP [5] is a combination of AHP with fuzzy set preference modeling, fuzzy multiple attribute decision-making methods [6], or other multiple criteria decision methods [7].

The most important functionality for a PLE is the process that adaptively provides a versatile course portfolio scheme for different learners from different perspectives. That is, the main objective of a PLE is to transform a learner from a role of passively accepting the course content organized by instructors, into a different role of proactively selecting the customized courses and contributing their knowledge to continuously improve the learning platform. The proposed scheme expedites the flexibility of lifelong learning according to the users' background and their requirements in a more flexible way.

We are aware that changing technologies are playing important roles as key drivers in educational change in the post-COVID-19 era. To make our discussion concise, this paper will not discuss the details regarding specific pedagogy (or didactics). Instead, we concentrate on the whole approach from a technological point of view and seek the possibilities of a contribution to the ecology of PLE.

In the following, we discuss related works on resolving the deficiencies of traditional e-learning and the inspiration of the e-learning 2.0 concept in Section 2. The basic elements of a general framework for PLE in an e-learning 2.0 environment will be addressed in Section 3. We elaborate a demonstrative example to explore the rationale of our approach in Section 4. In Section 5, we summarize our work and inspect some possibilities for future extension.

## 2. Related Works

On the road of working from e-learning 1.0 towards E-learning 2.0 [8,9], there are so many studies that have been conducted or proposed in the past decades. For example, Kundi & Nawaz (2014) [10] discuss the threats and opportunities for higher education institutions of shifting from traditional e-learning to e-learning 2.0, especially in developing countries. Wang & Chiu (2011) [11] explores the success factors of E-learning 2.0 systems and develops a theoretical model to assess user satisfaction and loyalty intentions to an e-learning system based on communication quality, information quality, system quality, and service quality.

Based on the rationale of Web 2.0, Huang & Shiu (2012) [12] proposes a user-centric E-learning 2.0 system (UALS), which conducts sequential pattern mining to construct adaptive learning paths to collect users' collective intelligence, and employs Item Response Theory (IRT) with collaborative voting to estimate learners' abilities for materials recommendation. By inviting learners to be "prosumers", Ferretti et al. (2008) [13] presents an E-learning 2.0 tool to support users in editing educational resources and compounding multimedia contents through collaborative work. Cristea & Ghali (2011) [14] study how to effectively

combine the peers' recommendation with content adaptation to enhance the learning outcome in an E-learning 2.0 environment.

In this study, based on a similar way of thinking, i.e., by synergetic merging Web 2.0, adaptation, and personalization into E-learning 2.0, we suggest separating the compound process into two stages, such that the upper stage is determined by learners' criteria (with AHP), and the lower stage is appraised by instructors.

To further explain the motivation of our work, we depict the concepts in Figures 1–4 to illustrate the drawbacks pointed out in Section 1. In Figure 1, instructors respectively offer their professional courses based on their subjective arrangements. Based on the traditional learning model in Figure 2, learners can only select the course packages organized by a single instructor with no flexibilities at all. To mitigate this shortcoming, we propose to divide a course into units (or topics) of high cohesion and coherence, and learners can customize their own course portfolios from different units offered by different instructors as shown in Figure 3.

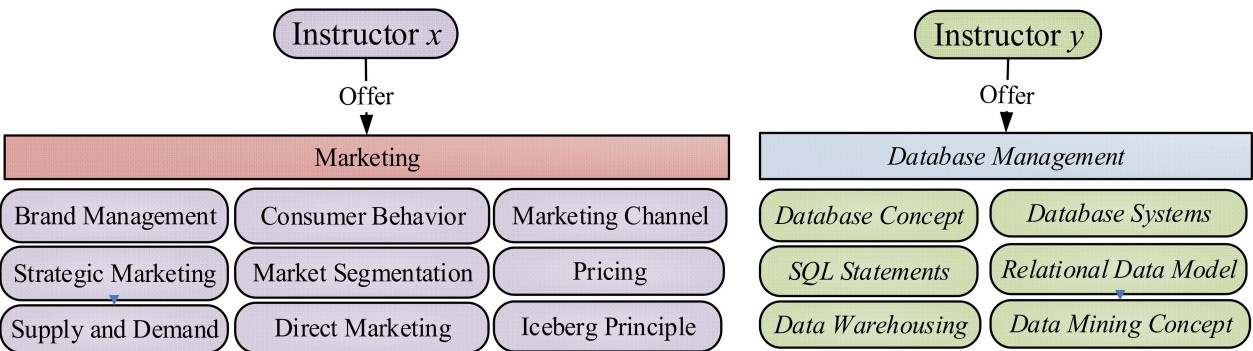

**Figure 1.** Traditional instructors offer courses by their specialties with subjective arrangements.

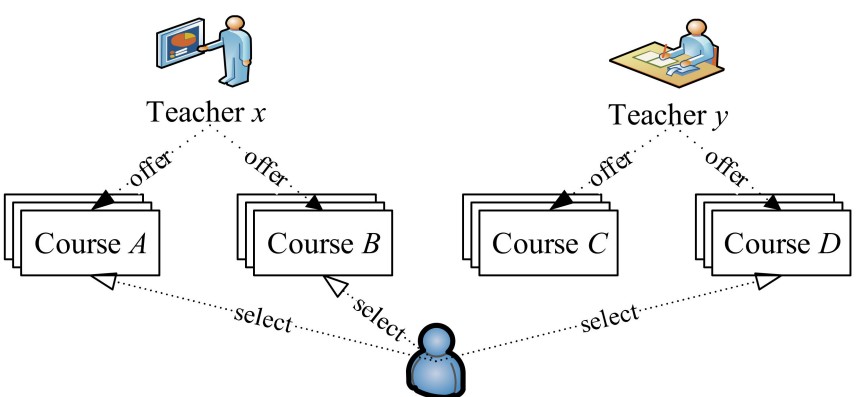

**Figure 2.** A scenario of traditional e-learning.

For example, suppose there are two instructors, and one offers a course in *marketing* and the other offers a course regarding *database management*. When a student *S* wants to learn how to apply database management technologies to his own business marketing applications, *S* has to join both courses to acquire the needed knowledge. If *S* does not have much time to learn both courses, then *S* may need to pay many times trying or searching for other acceptable alternatives to fulfill his demand. However, with the help of our approach with E-learning 2.0, *S* may proactively choose some of the topics in marketing and some of the topics in database management to customize his own course portfolio (it may be called *Database Marketing*) through the course portfolio mechanism as illustrated in Figure 4.

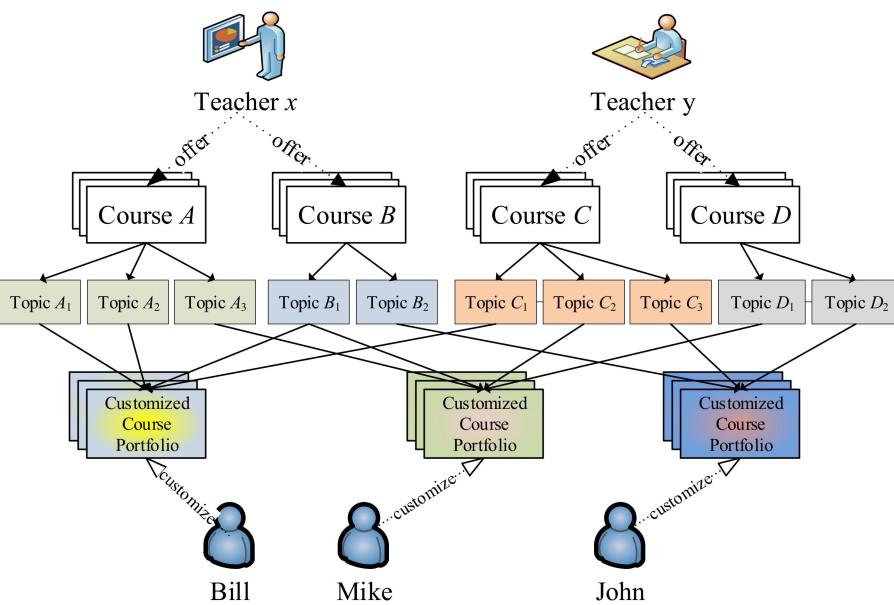

**Figure 3.** A scenario of E-learning 2.0.

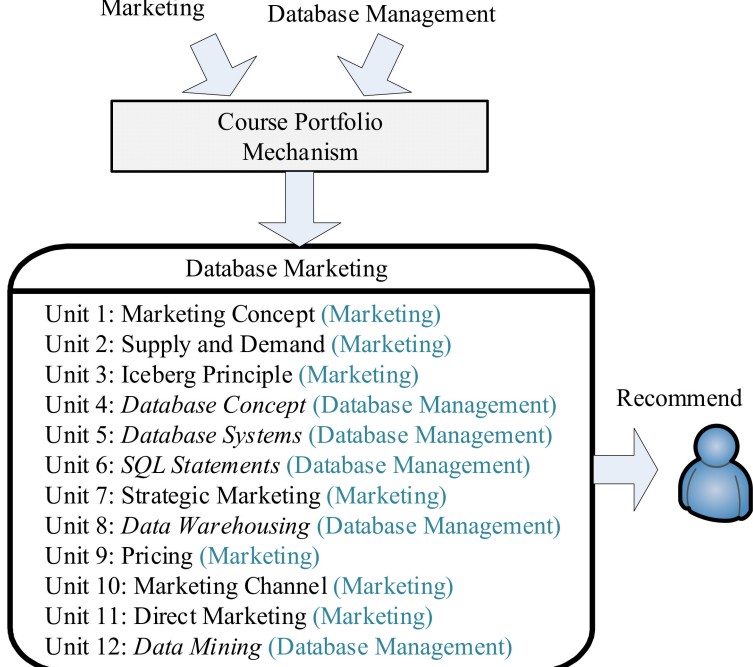

**Figure 4.** A customized course portfolio called *Database Marketing*.

Based on such rationale, the principle of designing the course portfolio mechanism for E-learning 2.0 is three-fold:

1. The mechanism should respect the profession of every instructor. That is, the intra-relationship between the units of a course and the inter-relationship between the units of different courses should be retained to some extent.
2. The mechanism needs to follow the requirements defined by the learner to generate acceptable results.
3. Although the topics in the derived course content are intermixed, they still have to retain their relative learning sequences defined by the instructors.

To achieve these objectives, we have proposed a general framework in [15] to meet these functionalities as illustrated in Figure 5. The detailed description of each module will be discussed in Section 4.

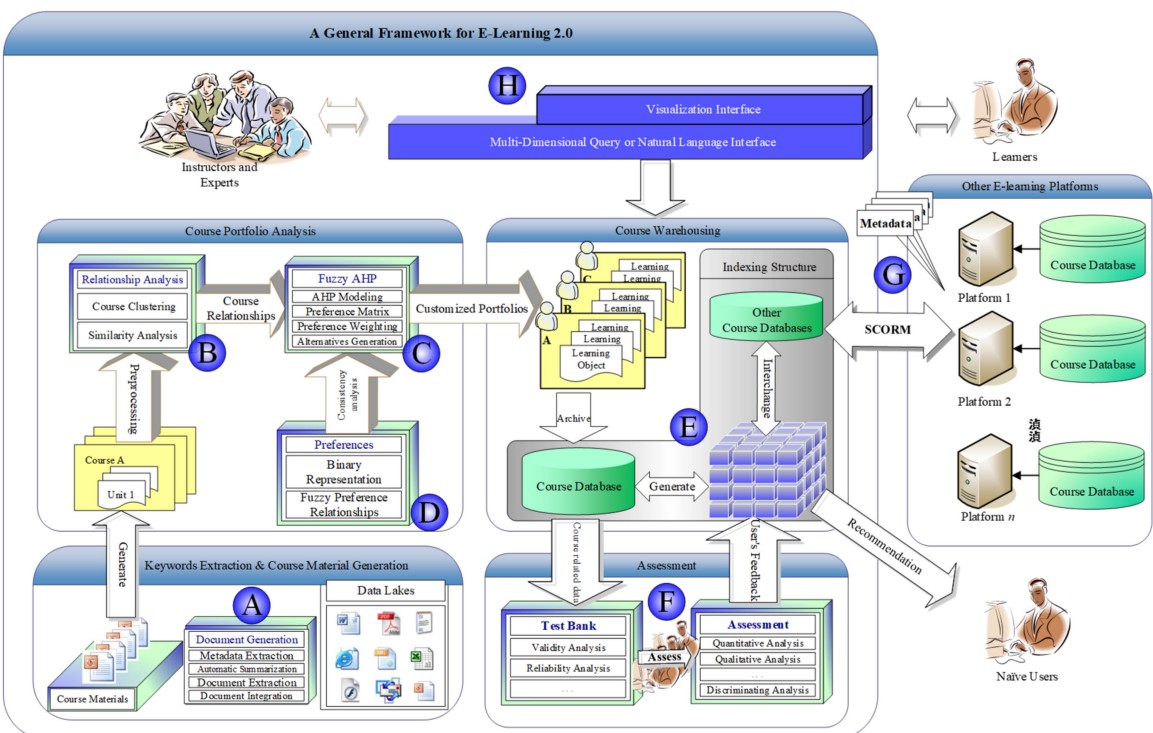

**Figure 5.** The proposed framework for E-Learning 2.0.

## 3. A General Framework for E-Learning 2.0

The proposed general framework in Figure 5 contains the following modules:

1.  *A Data Lake of Course Material*: Various types of course materials can be collected in this module. Each course material's metadata will be extracted, and course contents summarized [6,16,17], hyperlinked, and integrated with other course materials (as Node A indicates).
2.  *Automatic Course Portfolio Analysis*: Different courses composed of many units can be analyzed to find their relationships and similarities, and then clustered into many groups as node B specifies. This can be conducted by text mining processes in the background to help the system reduce the candidate courses when keywords are issued by a learner. By acquiring the necessary preferences input by users or instructors (node D), we can employ AHP (or Fuzzy AHP) to generate some of the alternatives containing the related learning objects for different learners (node C).
3.  *Course Warehousing*: To inspire an open e-learning system and offer hooks for integration with other systems, this module recommends using a multi-dimensional course warehousing structure to index the result obtained in the previous modules (node E). Based on such a flexible structure, the course content can be organized into hierarchies and pivoted for visualization from different perspectives. The course warehouse is only a multi-dimensional index pointing to the corresponding SCORM standard for further integration with other courses (node G).
4.  *Assessment*: By employing the multi-dimensional structure derived from the previous module, instructors can arrange two-way specification tables (like the example in Table 1) for a system to generate the test banks for the assessment of the corresponding courses (node F).
5.  *User Interfaces*: To provide a versatile user interface for learners or instructors to communicate with the system. Flexible visualization support is indispensable, as it

offers users (or instructors) to query the course database through a versatile multi-dimensional indexing structure (node H).

**Table 1.** A two-way specification table example.

| Knowledge ＼ Objectives | Remember | Understand | Apply | Analyze | Evaluate | Create | Sum |
|---|---|---|---|---|---|---|---|
| Factual Knowledge | 5 | 8 | 3 | 6 | 2 | 2 | 26 |
| Conceptual Knowledge | 8 | 10 | 4 | 5 | 4 | 2 | 33 |
| Procedural Knowledge | 5 | 3 | 2 | 3 | 4 | 2 | 19 |
| Meta-Cognitive Knowledge | 7 | 2 | 6 | 3 | 2 | 2 | 22 |
| *Sum* | 25 | 23 | 15 | 17 | 12 | 8 | 100 |

As indicated in [1], a PLE was not just an application software. It may consist of different apps used for learning in our daily life. Therefore, the framework in Figure 5 can be considered as a guideline to connect versatile apps to build an ecosystem for customized PLE. In the following, we discuss the mechanism (i.e., node C, the main focus of this study) for flexible course portfolio generation by orchestrating the learner and course design experts.

## 4. The Framework Based on Analytic Hierarchy Process

### 4.1. Preliminary

In our E-learning 2.0 framework, the basic building blocks are called learning objects, which may be formatted conforming to some kind of standard, like SCORM (Sharable Content Object Reference Model) [18,19], to avoid reinventing the wheel. That is, learning objects are the basic units, which can be combined in different ways for personalized learning portfolios. That makes learning objects be rigorously organized and packaged into courses for delivering knowledge from different perspectives. A successful personal e-learning system must integrate various learning objects and provide a platform for restructuring the building blocks as needed by different learners. Such a system, called a learning management system (LMS), can effectively manage and deliver online courses.

At the very least, a learning object should possess a suitable interface, and its content, together with some quality control metadata about version, date, content provider, manufacture, pre-requisites, to describe and index the learning object itself. For describing metadata, there are various well-established standards that can be applied, such as Dublin Core (1995) [20], IEEE Learning Object Metadata (2002) [21], or IMS Learning Resource Metadata Specification (2005) [22].

The knowledge design for a specific course could be mapped into a set of learning objects according to the famous Bloom's Taxonomy [23] or its revised counterparts [24,25] as depicted in Figure 6. In Bloom's Taxonomy, the *knowledge dimension* organizes domain knowledge into a hierarchy, which teaches users to learn about *what*, *how*, and then *why*, and the *cognitive process dimension* can be used as a guideline for designing the pedagogy for a specific course.

Finally, as the mapping between courses and learning objects is multi-dimensional in nature, we propose to organize the learning objects as a multi-dimensional document warehouse [26,27] to function as an intermediate communication layer for the content management of personalized interdisciplinary studies. With properly warehoused learning object structures, users can organize a course along some well-defined dimensions with related semantics. It not only provides a clear and guided pathway for organizing personalized courses, but also prevents exponential growth in the number of mappings as shown in Figure 7.

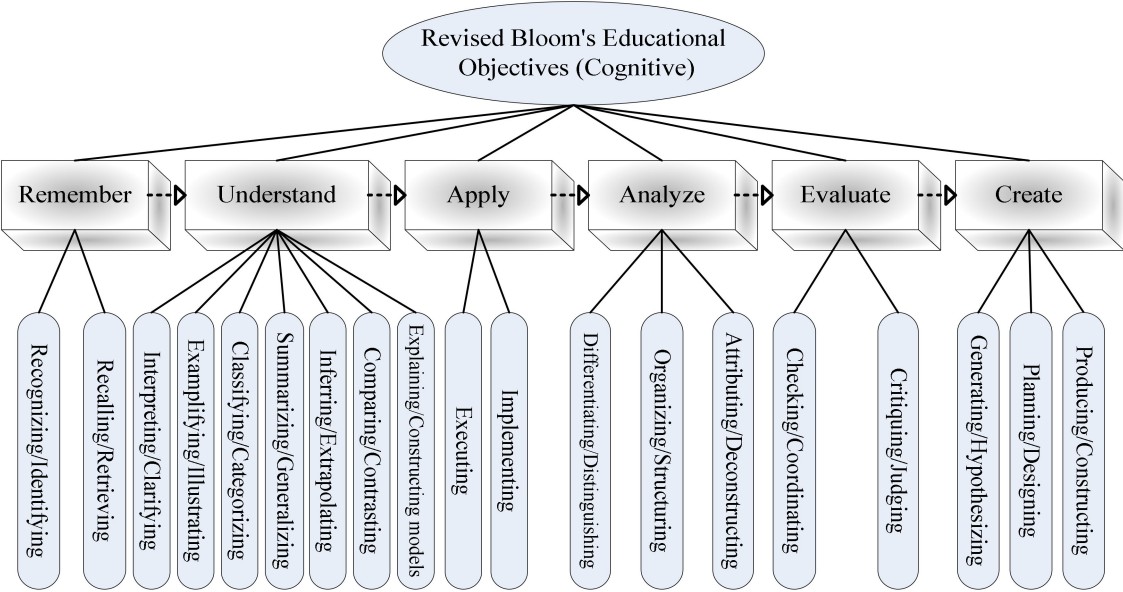

**Figure 6.** An illustration of revised Bloom's educational objectives.

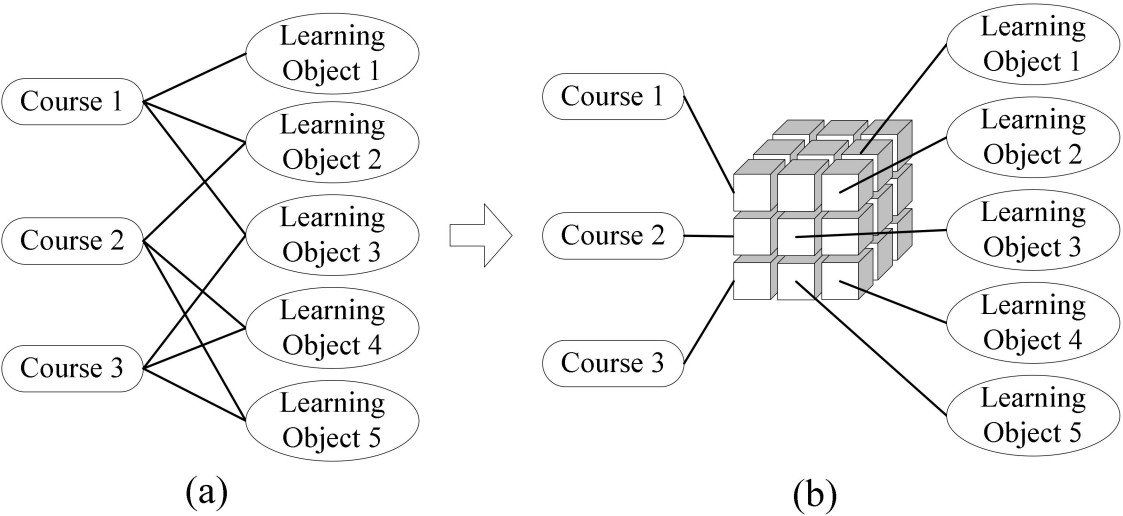

**Figure 7.** Organizing learning objects into a multi-dimensional topic cube. (**a**) The mapping between courses and learning objects are one-dimensional; however, (**b**) The mapping can be organized into a multi-dimensional structure.

To meet the principle of the course portfolio mechanism design, we propose a two-stage Analytic Hierarchy Process (AHP) for multi-criteria decision-making. It is characterized by systematically generating alternatives by quantified evaluation for multi-objective problems based on a hierarchical structure specified with multi-criteria. AHP is flexible and can be combined with other mechanisms, like Fuzzy Set Theory, to analyze or evaluate various applications [28] based on the following assumptions:

- The decision model can be decomposed into classes or components and organized into hierarchical and directed networks.
- All factors (objectives or criteria) are supposed to be independent.
- Use paired preference relationships to construct a comparison judgment matrix.
- Absolute scale can be transformed into ratio scale for judgments or evaluations.
- Preferences have transitivity.
- Non-transitive preferences are allowed to appear in the matrix.

- Taking into account all factors included in the hierarchy.

*4.2. The Proposed Two-Stage AHP Model*

Since the purpose of the final decision is to generate an intermixed course unit combination for a learner, namely *S*, the outer stage criteria should be set by *S*. However, as the unit relationships within a course should be determined by instructors (or experts), we design the decision model as a structure of two-stage AHP model as shown in Figure 8. That is, the outer stage concerns the learner side, and the inner stage corresponds to the instructors' side. The top two levels are defined by the learner, and the bottom two levels are defined by the instructors. We explain both stages bottom-up as follows:

1.  The inner stage: The purpose of this layer is to rank the units in each course. To achieve this goal, instructors have to define the pairwise intra-course comparisons of the associated units and define the pairwise inter-course scale ratios between the course and other courses. However, for *n* units, there are $n(n-1)/2$ comparisons that should be considered, and such tedious tasks may challenge the patience of instructors. Fortunately, the multiplicative transitivity property can be employed to reduce the complexity of the above dull task. For example, suppose there are *n* units, then instructors only need to set the $(n-1)$ pairwise comparisons of the entries above the matrix diagonal as follows:

$$R_{12} = u_1/u_2, R_{23} = u_2/u_3, ..., R_{(n-1)(n)} = u_{(n-1)}/u_{(n)} \qquad (1)$$

    Other ratio scales above the diagonal can be derived based on the transitivity property, e.g., $R_{13} = R_{12} \times R_{23} = u_1/u_2 \times u_2/u_3 = u_1/u_3$, $R_{24} = R_{23} \times R_{34} = u_2/u_3 \times u_3/u_4 = u_2/u_4$, ..., $R_{(n-2)(n)} = R_{(n-2)(n-1)} \times R_{(n-1)(n)} = u_{(n-2)}/u_{(n-1)} \times u_{(n-1)}/u_{(n)} = u_{(n-2)}/u_n$. Since $R_{ii} = 1$, for $1 \le i \le n$ and $R_{ij} = 1/R_{ji}$, the complete matrix can be obtained.

2.  The outer stage: Based on the criteria of learner *S*, the purpose of this layer is to generate the priorities of intermixed units selected from different courses to form a newly recommended course for *S*. To achieve this goal, the intra-course topic weights derived by the inner stage will be adopted. Suppose there are six factors (e.g., degree of difficulty (DoD), relationship, importance, sequence, bloom's taxonomy, and time constraints) can be set by *S*, which we explain as follows:

    (a) Degree of difficulty (DoD): It uses 1, 3, 5, 7, and 9 to respectively represent equal difficulty, weak difficulty, essential difficulty, very difficult and absolute difficulty; and use 2, 4, 6, 8 as the intermediate difficulties between the two adjacent judgments, respectively.

    (b) Relationship: It indicates the degree to which a unit can be learned independently; that is, the lower the connection with other units, the higher the chance that it can be disassembled. Its main purpose is to combine certain units that cannot be studied independently into a set to provide options for subsequent course combinations. The semantics of each scale of 1 to 9 are: absolutely irrelevant, very irrelevant, quite irrelevant, slightly irrelevant, equally relevant, slightly relevant, quite relevant, relevant, and extremely relevant.

    (c) Importance: This represents the importance of different courses, as well as the intra-course importance between units. Such criterion scales can be determined by instructors with different weights. The semantics of each scale of 1 to 9 are: absolutely unimportant, very unimportant, quite unimportant, slightly unimportant, equally important, slightly important, quite important, extremely important, and absolutely important.

    (d) Sequence: It denotes the order of dependency between certain units in a course. For example, learning JavaScript programming requires a basic knowledge of computer programming. Then, a computer programming unit of a course becomes the prerequisite of a JavaScript unit. The semantics of each scale of 1 to 9 are: must be behind, usually behind, often behind, sometimes behind,

not necessarily, sometimes in front, often in front, usually in front, and always in front.

    (e)    Bloom's Taxonomy: To specify a course unit with the six levels in Bloom's Taxonomy, where remember, understand, apply, analyze, evaluate, and create are represented by 1, 3, 5, 7, 8, 9, respectively.

    (f)    Time constraint: It is worth noting that the time constraint should not be considered as a pairwise comparison factor, as it is independent and can be calculated directly. The time constraint indicates the allocated time of *S* to learn the recommended course. If a user wants to learn certain course units only in a short time period, the system can set up this condition as a threshold for course selection, and thereby exclude courses or units that do not meet the time constraint by calculating the learning time of associated units.

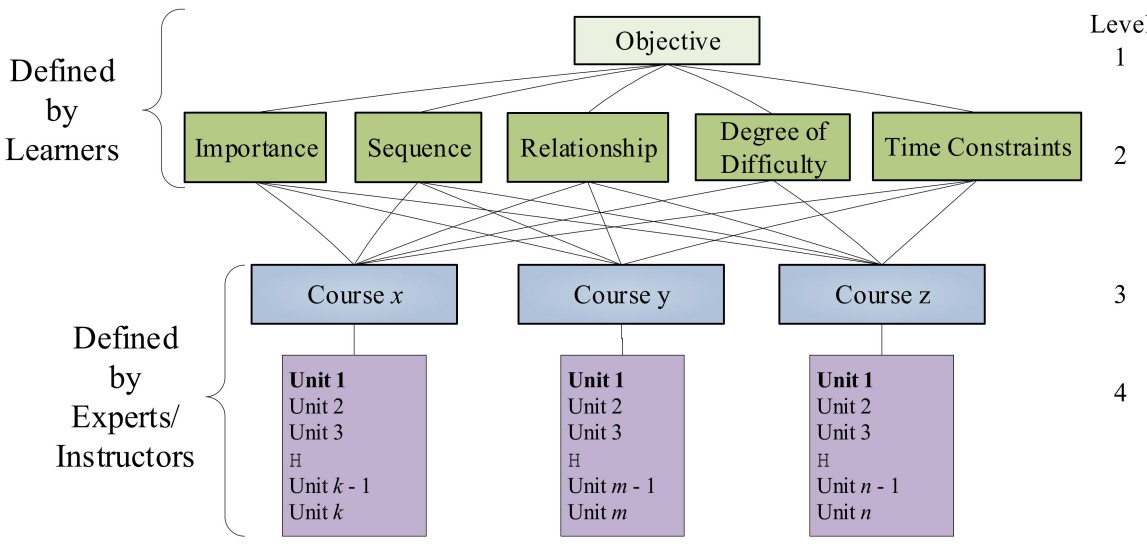

**Figure 8.** The proposed two-stage AHP framework.

Both stages of our AHP model can be generalized by propagating the result of inner AHP, which is conducted by an instructor-decided AHP, to the outer stage, evaluated by the learner-decided AHP, as illustrated by Figure 9. Each learner can select some predefined criteria (e.g., the importance, sequence, relationship, degree of difficulty, and time constraints) on the objective level, i.e., Level 2, using the available courses selected in Level 3, based on the derived intra-course weights in Level 4, which are in turn calculated from the criteria defined by instructors in Level 5, and applied to the units placed on Level 6.

In the evaluation process, each course can be regarded as a decision structure in the inner stage and is computed to derive the intra-course unit weights based on the instructor's criteria. Then, after all, selected courses are determined, the outer AHP is appraised to generate the customized course portfolio based on the learner's criteria settings. As such a process is tedious to explain, we use the simplest structure in Figure 8 to illustrate the concept.

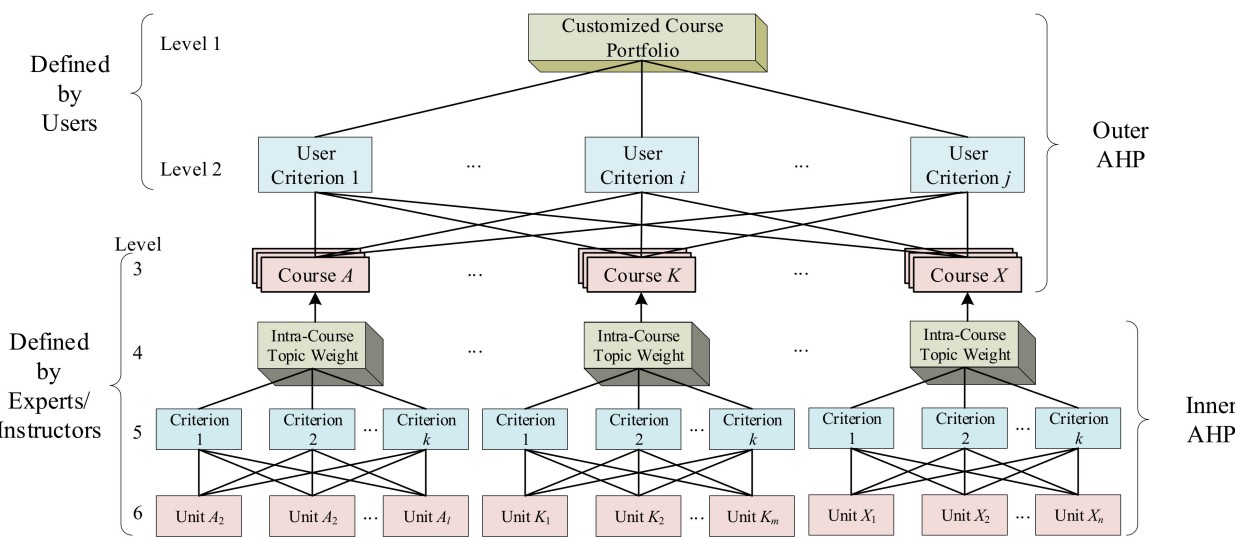

**Figure 9.** The Generalized two-stage AHP framework.

### 4.3. An Illustrative Example

In the following, we present an illustrative example to demonstrate our proposed concepts. Suppose learner *S* wants to allocate 20 hours for a customized course, intermixed with the courses of *JavaScript Programming*, *HTML* and *XML, and Internet Computing*, as described in Figure 10.

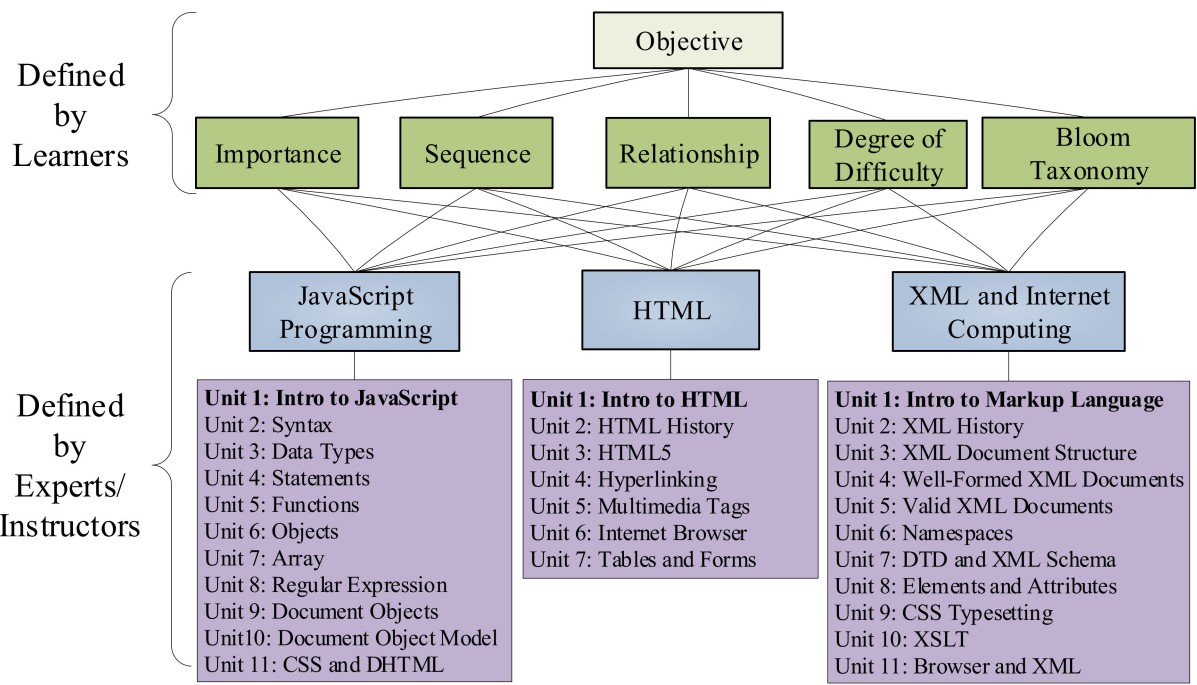

**Figure 10.** An example two-stage AHP illustration.

Suppose the pairwise comparison matrix of *JavaScript Programming* (denoted $a_i$) has been defined by the instructor in the second to sixth columns in Table 2. By applying the geometric normalization by means of the rows:

$$w_i = \frac{\sqrt[n]{\prod_{j=1}^{n} a_{ij}}}{\sum_{i=1}^{n} \prod_{j=1}^{n} a_{ij}}, \ i, j = 1, 2, ..., n \tag{2}$$

**Table 2.** The Pairwise Comparison Matrix for *JavaScript Programming*.

| | DoD | Relevance | Importance | Sequence | Bloom's | Average | Normalized Weight ($w$) | Priority Order | $w' = A \times w$ | $w'/w$ |
|---|---|---|---|---|---|---|---|---|---|---|
| DoD | 1 | 3 | 1 | 5 | 9 | 2.66727 | 0.3770 | 1 | 1.9700 | 5.22538 |
| Relevance | 1/3 | 1 | 1/3 | 1 | 7 | 0.95098 | 0.1344 | 3 | 0.6834 | 5.08535 |
| Importance | 1 | 3 | 1 | 3 | 9 | 2.40822 | 0.3403 | 2 | 1.7271 | 5.07438 |
| Sequence | 1/5 | 1 | 1/3 | 1 | 7 | 0.85862 | 0.1213 | 4 | 0.6332 | 5.21817 |
| Bloom Taxonomy | 1/9 | 1/7 | 1/9 | 1/7 | 1 | 0.19066 | 0.0270 | 5 | 0.1432 | 5.31370 |
| Sum | | | | | | 7.07576 | 1 | | | $\lambda_{\max}$ = 5.183396 |
| CR = CI/RI = 0.045849/1.12 = 0.040937 < 0.1 | | | | | | | | | | |

The related computation can be derived in the seventh to eighth columns in Table 2. We can derive the priority order from the vector in the ninth column as: Degree of Difficulty (DoD) > Importance > Relevance > Sequence > Bloom's Taxonomy. Besides, based on the derived $\lambda_{\max}$ = 5.183396, we can compute the Consistency Index, CI = $(\lambda_{\max} - n)/(n - 1)$ = (5.183396 − 5)/(5 − 1) = 0.183396/4 = 0.045849, and derive the Random Index, using RI = 1.12 by mapping to the Random Consistency table suggested by Saaty (1980) [2]. Then, verify the Consistency Ratio, CR = CI/RI = 0.045849/1.12 = 0.040937 < 0.1, which passes the consistency check. Similarly, we can apply the same process to *HTML* and *XML & Internet Computing* to derive their preference orders as listed in Tables 3 and 4, respectively. Their consistency ratios have been checked and listed below the tables, respectively.

We then move forward to the pairwise comparisons of the three courses on the third level, comparing them pairwise in satisfying each criterion on the second level. The involved instructors are responsible for this task. They need to generate five 3 × 3 matrices of judgments (since there are five elements on level two, and three courses to be pairwise compared for each element) through the interface provided by the system, as listed in Table 5. This is supposed to be done just once for each pair of courses.

The next step is to synthesize the priorities. By employing the distributive mode to establish the composite or global priorities of the courses, the instructors may need to provide the weighting values for all the course units, as shown in Tables 6 and 7.

If the instructors can patiently define the criteria weighting of their own course units (to be done just once), then the system can derive the fine-grained weighting of each unit and determine the unit priority orders as shown in Table 8.

However, assigning the weighting values for each course unit is tedious work. If the instructors are only responsible for assigning coarse-grained weights on the course level (instead of unit level), then the system can still use these course weights to compute the inter-course weight for each unit. For example, assume in Figure 11:

1.  The weights of course $x$ and $y$ are 0.4 and 0.6, respectively denoted $w(x) = 0.4$ and $w(y) = 0.6$,
2.  The weight of unit 1 of course $x$ is 0.3, denoted $w(x.1) = 0.3$,
3.  The weight of unit 1 of course $y$ is 0.2, denoted $w(y.1) = 0.2$,

**Table 3.** The Pairwise Comparison Matrix for *HTML*.

| | DoD | Relevance | Importance | Sequence | BloomTaxonomy | Average | Weight ($w$) | PriorityOrder | $w' = A \times w$ | $w'/w$ |
|---|---|---|---|---|---|---|---|---|---|---|
| DoD | 1 | 2 | 1 | 3 | 2 | 1.64375 | 0.2984495 | 1 | 1.6022 | 5.36858 |
| Relevance | 1/2 | 1 | 2 | 2 | 3 | 1.43097 | 0.2598154 | 2 | 1.3812 | 5.31601 |
| Importance | 1 | 1/2 | 1 | 2 | 3 | 1.24753 | 0.2261824 | 3 | 1.1743 | 5.19189 |
| Sequence | 1/3 | 1/2 | 1/2 | 1 | 2 | 0.69883 | 0.1268833 | 4 | 0.6467 | 5.09684 |
| Bloom Taxonomy | 1/2 | 1/3 | 1/3 | 1/2 | 1 | 0.48836 | 0.0886694 | 5 | 0.4633 | 5.22542 |
| Sum | | | | | | 5.50764 | 1 | | | $\lambda_{\max} = 5.239749$ |

CR = CI/RI = 0.059937/1.12 = 0.053515 < 0.1

**Table 4.** The Pairwise Comparison Matrix for *XML & Internet Computing*.

| | DoD | Relevance | Importance | Sequence | BloomTaxonomy | Average | Weight ($w$) | PriorityOrder | $w' = A \times w$ | $w'/w$ |
|---|---|---|---|---|---|---|---|---|---|---|
| DoD | 1 | 5 | 1 | 7 | 9 | 3.15982 | 0.4258477 | 1 | 2.3175 | 5.44218 |
| Relevance | 1/5 | 1 | 1/3 | 1 | 7 | 0.85862 | 0.1157160 | 3 | 0.5971 | 5.16025 |
| Importance | 1 | 3 | 1 | 3 | 9 | 2.40822 | 0.3245556 | 2 | 1.6533 | 5.09424 |
| Sequence | 1/7 | 1 | 1/3 | 1 | 7 | 0.80274 | 0.1081852 | 4 | 0.5728 | 5.29453 |
| Bloom Taxonomy | 1/9 | 1/7 | 1/9 | 1/7 | 1 | 0.19066 | 0.0256954 | 5 | 0.1411 | 5.48967 |
| Sum | | | | | | 7.42007 | 1 | | | $\lambda_{\max} = 5.296175$ |

CR = CI/RI = 0.045849/1.12 = 0.040937 < 0.1

Then the inter-course weight of unit 1 of course *x* is 0.3 × 0.4, denoted *icw*(*x*.1) = 0.12, and the inter-course weight of unit 1 of course *y* is 0.2 × 0.6, denoted *icw*(*y*.1) = 0.12.

Assume the courses are assigned with weights (denoted $W_C$) and then normalized (denoted $N_C$) as listed in Table 9.

Then, the system can derive the weights for all units of the courses by multiply $J_U$ with the transpose of $N_C$[*JavaScript Programming*] (i.e., $J_U \times N_C$[*JavaScript Programming*]$^T$), $H_U$ with the transpose of $N_C$[*HTML*] (i.e., $H_U \times N_C$[*HTML*]$^T$) and $X_U$ with the transpose of $N_C$[*XML*] (i.e., $X_U \times N_C$[*XML*]$^T$) to produce the result shown in Table 10.

Then, the evaluation of the unit weights for these courses can be elaborated in Table 11.

Finally, by assuming it costs one hour to learn each unit, and as the learner has only 20 h for studying the combined course, some of the course units ordered after 20 will be discarded. The final customized course can be organized by the original sequences in the courses as Table 12 illustrates.

That is, by arranging the sequences of units in the corresponding courses, the final course units recommended to the user can be listed in Table 13.

**Table 5.** The generated five 3 × 3 matrices of judgments for *JavaScript Programming*, *HTML*, and *XML & Internet Computing*.

| | *JavaScript Programming* | *HTML* | *XML & Internet Computing* | **Normalized Priorities** |
|---|---|---|---|---|
| *DoD* [1] | | | | |
| *JavaScript* | 1 | 6 | 9 | 0.762634 |
| *HTML* | 1/6 | 1 | 4 | 0.17626 |
| *XML & Internet Computing* | 1/9 | 1/4 | 1 | 0.061106 |
| *Relevance* [2] | | | | |
| *JavaScript* | 1 | 4 | 4 | 0.673811 |
| *HTML* | 1/5 | 1 | 1/3 | 0.100654 |
| *XML & Internet Computing* | 1/4 | 1/3 | 1 | 0.225535 |
| *Importance* [3] | | | | |
| *JavaScript* | 1 | 5 | 7 | 0.730645 |
| *HTML* | 1/5 | 1 | 4 | 0.188394 |
| *XML & Internet Computing* | 1/7 | 1/3 | 1 | 0.080961 |
| *Sequence* [4] | | | | |
| *JavaScript* | 1 | 3 | 6 | 0.66667 |
| *HTML* | 1/3 | 1 | 2 | 0.22222 |
| *XML & Internet Computing* | 1/6 | 1/2 | 1 | 0.11111 |
| *Bloom's* [5] | | | | |
| *JavaScript* | 1 | 1/5 | 1/4 | 0.093616 |
| *HTML* | 5 | 1 | 3 | 0.626696 |
| *XML & Internet Computing* | 4 | 1/3 | 1 | 0.279688 |

[1] $\lambda_{max}$ = 3.107847, C.I. = 0.053924, C.R. = 0.092972; [2] $\lambda_{max}$ = 3.085767, C.I. = 0.042883, C.R. = 0.073937; [3] $\lambda_{max}$ = 3.064888, C.I. = 0.032444, C.R. = 0.055938; [4] $\lambda_{max}$ = 3.000000, C.I. = 0.000000, C.R. = 0.000000; [5] $\lambda_{max}$ = 3.085767, C.I. = 0.042883, C.R. = 0.073937.

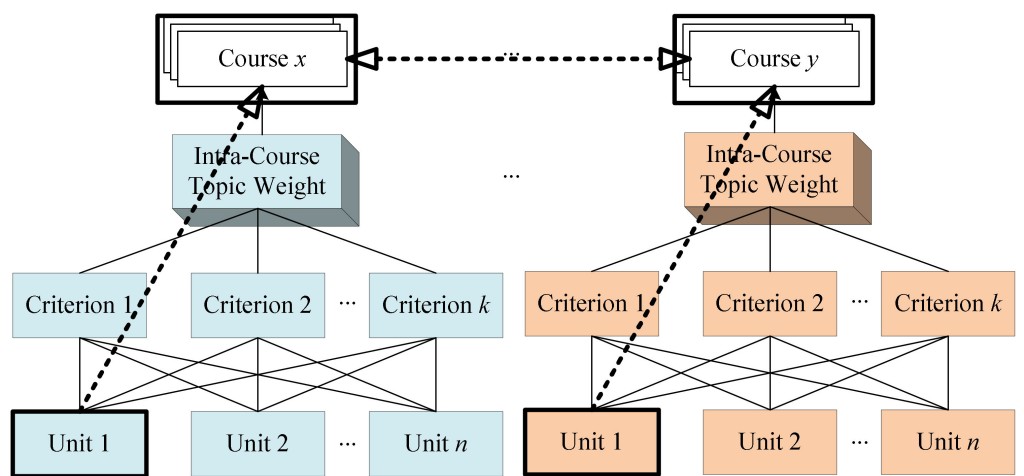

**Figure 11.** Inter-Course Weighting for Each Unit.

**Table 6.** The weighting values of the criteria for the units of *JavaScript Programming* and *XML & Internet Computing*.

| | JavaScript Programming ($J_U$) | | | | | XML and Internet Computing ($X_U$) | | | | |
|---|---|---|---|---|---|---|---|---|---|---|
| | DoD | Relevance | Importance | Sequence | Bloom's | DoD | Relevance | Importance | Sequence | Bloom's |
| Unit 1 | 0.003 | 0.086 | 0.192 | 0.047 | 0.108 | 0.031 | 0.144 | 0.047 | 0.112 | 0.102 |
| Unit 2 | 0.154 | 0.090 | 0.098 | 0.051 | 0.041 | 0.045 | 0.038 | 0.081 | 0.029 | 0.061 |
| Unit 3 | 0.096 | 0.137 | 0.030 | 0.215 | 0.225 | 0.110 | 0.034 | 0.106 | 0.234 | 0.111 |
| Unit 4 | 0.015 | 0.077 | 0.110 | 0.053 | 0.178 | 0.096 | 0.085 | 0.079 | 0.054 | 0.100 |
| Unit 5 | 0.190 | 0.005 | 0.053 | 0.168 | 0.118 | 0.169 | 0.199 | 0.192 | 0.173 | 0.044 |
| Unit 6 | 0.169 | 0.042 | 0.085 | 0.217 | 0.008 | 0.123 | 0.119 | 0.154 | 0.002 | 0.226 |
| Unit 7 | 0.071 | 0.155 | 0.174 | 0.079 | 0.282 | 0.069 | 0.186 | 0.066 | 0.204 | 0.068 |
| Unit 8 | 0.122 | 0.154 | 0.144 | 0.141 | 0.010 | 0.229 | 0.091 | 0.167 | 0.025 | 0.096 |
| Unit 9 | 0.179 | 0.255 | 0.114 | 0.030 | 0.029 | 0.128 | 0.104 | 0.108 | 0.166 | 0.191 |
| Unit 10 | 0.046 | 0.028 | 0.095 | 0.134 | 0.075 | 0.214 | 0.111 | 0.049 | 0.245 | 0.221 |
| Unit 11 | 0.031 | 0.134 | 0.036 | 0.192 | 0.092 | 0.139 | 0.213 | 0.064 | 0.289 | 0.052 |

**Table 7.** The weighting values of the criteria for the units of *HTML*.

| | HTML ($H_U$) | | | | |
|---|---|---|---|---|---|
| | DoD | Relevance | Importance | Sequence | Bloom's |
| Unit 1 | 0.003 | 0.086 | 0.192 | 0.047 | 0.108 |
| Unit 2 | 0.11 | 0.090 | 0.098 | 0.051 | 0.041 |
| Unit 3 | 0.26 | 0.137 | 0.030 | 0.215 | 0.225 |
| Unit 4 | 0.015 | 0.077 | 0.110 | 0.053 | 0.178 |
| Unit 5 | 0.190 | 0.005 | 0.053 | 0.168 | 0.118 |
| Unit 6 | 0.169 | 0.042 | 0.085 | 0.217 | 0.008 |
| Unit 7 | 0.071 | 0.155 | 0.174 | 0.079 | 0.282 |

**Table 8.** Determine the priority order and weights of the units in *JavaScript*.

| | JavaScript Programming | | | | | | Course Weighting of JavaScript Programming | | | Unit Weighting | |
|---|---|---|---|---|---|---|---|---|---|---|---|
| | DoD | Relevance | Importance | Sequence | Bloom | | | | | Weights | Order |
| Unit 1 | 0.003 | 0.086 | 0.192 | 0.047 | 0.108 | | DoD | 0.2836 | | 0.103 | 5 |
| Unit 2 | 0.154 | 0.090 | 0.098 | 0.051 | 0.041 | | Relevance | 0.1763 | | 0.089 | 8 |
| Unit 3 | 0.096 | 0.137 | 0.030 | 0.215 | 0.225 | | Importance | 0.3324 | | 0.128 | 2 |
| Unit 4 | 0.015 | 0.077 | 0.110 | 0.053 | 0.178 | × | Sequence | 0.1763 | = | 0.103 | 6 |
| Unit 5 | 0.190 | 0.005 | 0.053 | 0.168 | 0.118 | | Timing | 0.0315 | | 0.105 | 4 |
| Unit 6 | 0.169 | 0.042 | 0.085 | 0.217 | 0.008 | | | | | 0.083 | 9 |
| Unit 7 | 0.071 | 0.155 | 0.174 | 0.079 | 0.282 | | | | | 0.177 | 1 |
| Unit 8 | 0.122 | 0.154 | 0.144 | 0.141 | 0.010 | | | | | 0.098 | 7 |
| Unit 9 | 0.179 | 0.255 | 0.114 | 0.030 | 0.029 | | | | | 0.112 | 3 |
| Unit 10 | 0.046 | 0.028 | 0.095 | 0.134 | 0.075 | | | | | 0.073 | 10 |
| Unit 11 | 0.031 | 0.134 | 0.036 | 0.192 | 0.092 | | | | | 0.072 | 11 |

**Table 9.** Different weights of different criteria can be assigned for different courses.

| $W_C$ | | | | | |
|---|---|---|---|---|---|
| *Level 6* | *DoD* | *Relevance* | *Importance* | *Sequence* | *Bloom's Taxonomy* |
| *JavaScript* | 0.7 | 0.45 | 0.45 | 0.3 | 0.7 |
| *HTML* | 0.2 | 0.35 | 0.35 | 0.5 | 0.1 |
| *XML* | 0.1 | 0.2 | 0.2 | 0.2 | 0.2 |
| Normalize | | ↓ | | | |
| $N_C$ | | | | | |
| *Level 6* | *DoD* | *Relevance* | *Importance* | *Sequence* | *Bloom's Taxonomy* |
| *JavaScript* | 0.269230769 | 0.173076923 | 0.173076923 | 0.115384615 | 0.269230769 |
| *HTML* | 0.133333333 | 0.233333333 | 0.233333333 | 0.333333333 | 0.066666667 |
| *XML* | 0.111111111 | 0.222222222 | 0.222222222 | 0.222222222 | 0.222222222 |

**Table 10.** Different weights of different criteria can be assigned for different courses.

| $W_U$ | | | | | |
|---|---|---|---|---|---|
| *JavaScript Programming* | *Weight* | *HTML* | *Weight* | *XML and Internet Computing* | *Weight* |
| Unit 1 | 0.021819527 | Unit 1 | 0.012111111 | Unit 1 | 0.015740741 |
| Unit 2 | 0.017455621 | Unit 2 | 0.026644444 | Unit 2 | 0.012382716 |
| Unit 3 | 0.026183432 | Unit 3 | 0.035122222 | Unit 3 | 0.022666667 |
| Unit 4 | 0.021819527 | Unit 4 | 0.039482222 | Unit 4 | 0.014271605 |
| Unit 5 | 0.021819527 | Unit 5 | 0.042388889 | Unit 5 | 0.02308642 |
| Unit 6 | 0.017455621 | Unit 6 | 0.0436 | Unit 6 | 0.025185185 |
| Unit 7 | 0.026183432 | Unit 7 | 0.042873333 | Unit 7 | 0.010493827 |
| Unit 8 | 0.02072855 | | | Unit 8 | 0.02308642 |
| Unit 9 | 0.021819527 | | | Unit 9 | 0.020987654 |
| Unit 10 | 0.011564349 | | | Unit 10 | 0.020987654 |
| Unit 11 | 0.011346154 | | | Unit 11 | 0.020987654 |

**Table 11.** Different weights of different criteria can be assigned for different courses.

| | *JavaScript Programming* | | | | | Course Weighting of *JavaScript Programming* | | Unit Weighting | |
|---|---|---|---|---|---|---|---|---|---|
| | *DoD* | *Relevance* | *Importance* | *Sequence* | *Bloom* | | | Weights | Order |
| Unit 1 | 0.026923077 | 0.017307692 | 0.017307692 | 0.011538462 | 0.026923077 | DoD | 0.269230769 | 0.021819527 | 3 |
| Unit 2 | 0.021538462 | 0.013846154 | 0.013846154 | 0.009230769 | 0.021538462 | Relevance | 0.173076923 | 0.017455621 | 8 |
| Unit 3 | 0.032307692 | 0.020769231 | 0.020769231 | 0.013846154 | 0.032307692 | Importance | 0.173076923 | 0.026183432 | 1 |
| Unit 4 | 0.026923077 | 0.017307692 | 0.017307692 | 0.011538462 | 0.026923077 × | Sequence | 0.115384615 = | 0.021819527 | 4 |
| Unit 5 | 0.026923077 | 0.017307692 | 0.017307692 | 0.011538462 | 0.026923077 | Timing | 0.269230769 | 0.021819527 | 5 |
| Unit 6 | 0.021538462 | 0.013846154 | 0.013846154 | 0.009230769 | 0.021538462 | | | 0.017455621 | 9 |
| Unit 7 | 0.032307692 | 0.020769231 | 0.020769231 | 0.013846154 | 0.032307692 | | | 0.026183432 | 2 |
| Unit 8 | 0.025576923 | 0.016442308 | 0.016442308 | 0.010961538 | 0.025576923 | | | 0.02072855 | 7 |
| Unit 9 | 0.026923077 | 0.017307692 | 0.017307692 | 0.011538462 | 0.026923077 | | | 0.021819527 | 6 |
| Unit 10 | 0.014269231 | 0.009173077 | 0.009173077 | 0.006115385 | 0.014269231 | | | 0.011564349 | 10 |
| Unit 11 | 0.014 | 0.009 | 0.009 | 0.006 | 0.014 | | | 0.011346154 | 11 |

**Table 11.** *Cont*.

| | HTML | | | | | Course Weighting of HTML | | Unit Weighting | |
|---|---|---|---|---|---|---|---|---|---|
| | *DoD* | *Relevance* | *Importance* | *Sequence* | *Bloom* | | | Weights | Order |
| Unit 1 | 0.006666667 | 0.011666667 | 0.011666667 | 0.016666667 | 0.003333333 | DoD | 0.133333333 | 0.012111111 | 7 |
| Unit 2 | 0.014666667 | 0.025666667 | 0.025666667 | 0.036666667 | 0.007333333 | Relevance | 0.233333333 | 0.026644444 | 6 |
| Unit 3 | 0.019333333 | 0.033833333 | 0.033833333 | 0.048333333 | 0.009666667 | Importance | 0.233333333 | 0.035122222 | 5 |
| Unit 4 | 0.021733333 | 0.038033333 | 0.038033333 | 0.054333333 | 0.010866667× | Sequence | 0.333333333 = | 0.039482222 | 4 |
| Unit 5 | 0.023333333 | 0.040833333 | 0.040833333 | 0.058333333 | 0.011666667 | Timing | 0.066666667 | 0.042388889 | 3 |
| Unit 6 | 0.024 | 0.042 | 0.042 | 0.06 | 0.012 | | | 0.0436 | 1 |
| Unit 7 | 0.0236 | 0.0413 | 0.0413 | 0.059 | 0.0118 | | | 0.042873333 | 2 |
| | *XML and Internet Computing* | | | | | Course Weighting of *XML & Internet Computing* | | Unit Weighting | |
| | *DoD* | *Relevance* | *Importance* | *Sequence* | *Bloom* | | | Weights | Order |
| Unit 1 | 0.008333333 | 0.016666667 | 0.016666667 | 0.016666667 | 0.016666667 | DoD | 0.111111111 | 0.015740741 | 8 |
| Unit 2 | 0.006555556 | 0.013111111 | 0.013111111 | 0.013111111 | 0.013111111 | Relevance | 0.222222222 | 0.012382716 | 10 |
| Unit 3 | 0.012 | 0.024 | 0.024 | 0.024 | 0.024 | Importance | 0.222222222 | 0.022666667 | 4 |
| Unit 4 | 0.007555556 | 0.015111111 | 0.015111111 | 0.015111111 | 0.015111111× | Sequence | 0.222222222 = | 0.014271605 | 9 |
| Unit 5 | 0.012222222 | 0.024444444 | 0.024444444 | 0.024444444 | 0.024444444 | Timing | 0.222222222 | 0.02308642 | 2 |
| Unit 6 | 0.013333333 | 0.026666667 | 0.026666667 | 0.026666667 | 0.026666667 | | | 0.025185185 | 1 |
| Unit 7 | 0.005555556 | 0.011111111 | 0.011111111 | 0.011111111 | 0.011111111 | | | 0.010493827 | 11 |
| Unit 8 | 0.012222222 | 0.024444444 | 0.024444444 | 0.024444444 | 0.024444444 | | | 0.02308642 | 3 |
| Unit 9 | 0.011111111 | 0.022222222 | 0.022222222 | 0.022222222 | 0.022222222 | | | 0.020987654 | 5 |
| Unit 10 | 0.011111111 | 0.022222222 | 0.022222222 | 0.022222222 | 0.022222222 | | | 0.020987654 | 6 |
| Unit 11 | 0.011111111 | 0.022222222 | 0.022222222 | 0.022222222 | 0.022222222 | | | 0.020987654 | 7 |

**Table 12.** Global Priority Ordering to Determine the Customized Course.

| | | | $Wu$ | | | | | |
|---|---|---|---|---|---|---|---|---|
| Course.Unit | Weight | Order | Course.Unit | Weight | Order | Course.Unit | Weight | Order |
| HTML.U6 | 0.0436 | 1 | XML.U3 | 0.022666667 | 12 | XML.U11 | 0.020988 | 19 |
| HTML.U7 | 0.042873333 | 2 | JavaScript.U1 | 0.021819527 | 13 | JavaScript.U8 | 0.020729 | 20 |
| HTML.U5 | 0.042388889 | 3 | JavaScript.U4 | 0.021819527 | 14 | ~~JavaScript.U2~~ | ~~0.017456~~ | ~~21~~ |
| HTML.U4 | 0.039482222 | 4 | JavaScript.U5 | 0.021819527 | 15 | ~~JavaScript.U6~~ | ~~0.017456~~ | ~~22~~ |
| HTML.U3 | 0.035122222 | 5 | JavaScript.U9 | 0.021819527 | 16 | ~~XML.U1~~ | ~~0.015741~~ | ~~23~~ |
| HTML.U2 | 0.026644444 | 6 | XML.U9 | 0.020987654 | 17 | ~~XML.U4~~ | ~~0.014272~~ | ~~24~~ |
| JavaScript.U3 | 0.026183432 | 7 | XML.U10 | 0.020987654 | 18 | ~~XML.U2~~ | ~~0.012383~~ | ~~25~~ |
| JavaScript.U7 | 0.026183432 | 8 | | | | ~~HTML.U1~~ | ~~0.012111~~ | ~~26~~ |
| XML.U6 | 0.025185185 | 9 | | | | ~~JavaScript.U10~~ | ~~0.011564~~ | ~~27~~ |
| XML.U5 | 0.02308642 | 10 | | | | ~~JavaScript.U11~~ | ~~0.011346~~ | ~~28~~ |
| XML.U8 | 0.02308642 | 11 | | | | ~~XML.U7~~ | ~~0.010494~~ | ~~29~~ |

**Table 13.** Global Priority Ordering to Determine the Customized Course.

| Recommended Course Unit (1 to 10) | Recommended Course Unit (11 to 20) |
| --- | --- |
| HTML.U2: *HTML History* | JavaScript.U7: *Array* |
| HTML.U3: *HTML5* | JavaScript.U8: *Regular Expression* |
| HTML.U4: *Hyperlinking* | JavaScript.U9: *Document Objects* |
| HTML.U5: *Multimedia Tags* | XML.U3: *XML Doc Structure* |
| HTML.U6: *Internet Browser* | XML.U5: *Valid XML Documents* |
| HTML.U7: *Tables and Forms* | XML.U6: *Namespaces* |
| JavaScript.U1: *Introduction to JavaScript* | XML.U8: *Elements and Attributes* |
| JavaScript.U3: *Data Types* | XML.U9: *CSS Typesetting* |
| JavaScript.U4: *Statements* | XML.U10: *XSLT* |
| JavaScript.U5: *Functions* | XML.U11: *Browser and XML* |

## 5. Conclusions

Personalized e-learning is about studying professional curriculums through the Internet with some personal considerations. As lifelong learning and Internet-based education begin to emerge and proliferate, the need for customizing educational materials becomes increasingly important. The issues discussed in this paper attempt to integrate the computerized learning objects and create specifications that allow multiple instructors to work together to develop valuable and reconfigurable learning content. By incorporating the application of professional principles of learning/education and appropriate instructional design, such a learning environment can support extremely high quality, student-centered education programs for remote learners, making extensive use of the synchronous and asynchronous tools available for Internet-based communications.

Today's instructors also face some tensions from professional development, which are gradually amplified as new technologies are used to provide e-learning. Therefore, they also need to pay more engagement with new ideas, to involve varied enactment in practice, to rethink their roles and identities, and to change interaction with the world outside their classroom [29].

The approach is a highly interactive and efficient environment for instructors and learners, such that generated learning materials can be gradually evolved through learners themselves and be adapted by stakeholders as a foundation for underpinning the concept of learning ecologies for lifelong learning as discussed in [30].

We believe that a successful e-learning environment has to integrate various topics and provide a platform for restructuring the building blocks as needed by different learners. Besides, a fruitful e-learning infrastructure should be armed with the ability to absorb users' learning experiences and utilize that information to recommend casual users based on their background and requirements.

We realize that no matter how wonderful the mechanism a system adopts, it cannot do much without a good content organization of the domain on which it is to work. Moreover, we often recognize that, once a good content organization is available, many different mechanisms might be employed equally well to implement effective systems. Therefore, we claim that the ultimate solution is to provide a flexible course content management framework for learners to dynamically customize their course contents.

While data warehouses and numeric-centric business intelligence technologies have served most enterprises well, they do not fully address the complete scope of business intelligence. In this paper, we advocate the importance of indexing learning objects into document warehouses to support text-centric business intelligence and propose the architecture for the next generation e-learning environment. When learning objects are properly warehoused, users can perform ad hoc online analytical processing (OLAP) over course materials in a structured micro-context, just as the way users can perform OLAP over

summarized data in a data warehouse. Besides, users can customize their needed courses according to the dimensions of a topic cube easily for interdisciplinary study.

The concept of document warehousing is not only providing the ability to very fast learning object access without degradation in performance even as the size of the cube grows, but also offering a set of versatile applications for content management of e-learning and enterprise business intelligence.

When learning objects are properly warehoused, the task of version control will become very easy, since users can directly trace the topics based on some criteria along the time dimension. Such merits also make document warehousing an exhilarating organization for online topic detecting and event tracking on users' learning [31]. Besides, learning object clustering can be achieved directly via visualizations on a cube. Users can also develop some summarization tools [6,17,32] to summarize a cluster of related learning objects. To sum up, our approach is not only one of the best infrastructures for content management in e-learning, but also supports a flexible personalized learning environment. In the near future, we will elaborate on exploring more techniques to implement this framework, conducting some experiments on the testbed, and creating a learning ecology in distributed environments [33].

**Author Contributions:** Conceptualization, F.S.C.T. and C.-T.Y.; methodology, F.S.C.T. and C.-T.Y.; software, C.-T.Y.; validation, F.S.C.T., C.-T.Y. and A.Y.H.C.; formal analysis, F.S.C.T.; investigation, C.-T.Y.; resources, F.S.C.T.; data curation, F.S.C.T.; writing—original draft preparation, F.S.C.T.; writing—review and editing, F.S.C.T. and A.Y.H.C.; visualization, F.S.C.T. and C.-T.Y.; supervision, F.S.C.T.; project administration, F.S.C.T.; funding acquisition, F.S.C.T. All authors have read and agreed to the published version of the manuscript.

**Funding:** This research was funded by the Ministry of Science and Technology, Taiwan, ROC, under Contract No. MOST 109-2410-H-992-023-MY2.

**Acknowledgments:** This research was partially supported by the Ministry of Science and Technology, Taiwan, ROC, under Contract No. MOST 109-2410-H-992-023-MY2.

**Conflicts of Interest:** The authors declare no conflict of interest.

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
