# Peer review of "A Collaborative Framework for Customized E-Learning Services by Analytic Hierarchy Processing"

_applsci, doi:10.3390/app12031377_

Round 1

Reviewer 1 Report

 A Collaborative Framework for Customized E-Learning Services by Analytic Hierarchy Processing  

The authors point out that the main objective of our framework is to transform a learner from a role of passively accepting the course content organized by instructors, into another role of proactively selecting the courses and contributing their knowledge to continuously improve the learning platform. We believe the approach proposed is a versatile way for supporting various challenges for next generation e-learning environment.

 The objectives are well set, but the whole article works somewhat inconsistently. I would like to make some general comments and questions below and some particular comments.

General comments:

  1. Try to initially display and explain the entire "system" and then explain the individual elements below. As a reader and reviewer, I didn't see the whole and I found it very difficult to get through the article. The article is educational, so you should also explain the pedagogical basics as,
  • some kind of pedagogy embedded in the learning process - which one, more concrete!!

- By incorporating the application of professional pedagogical principles of learning/education and appropriate instructionaldesign - which one, more concrete!!  mentioned only in Conclusion

  1. In Abstract and Introduction lacks a comprehensive explanation of the base idea. e.g.: Abstract: To whom this system is intended (target groups), function and composition, etc. The technological part is well defined, but the "education part" is completely undefined.
  2. E-learning - better in your case distance learning

  1. Throughout the course, you mix customized learning platforms - Course portfolio - learning environment! Be consistent in the article because with different terms for "same"? thing you are misleading the reader!

  1. Role of instructors, learners and system? - Additional explanation in abstract and introduction.
  2. Correlations instructors - learners - system are extremely poorly defined. You use the word "we" - who is it, developers, instructors, system?? In An Illustrative Example Who and what - we have learners, instructors and I supposed system? What mean "we"??
  3. Each learner can define their criteria? (e.g., the importance, sequence, relationship, degree of difficulty, and time constraints) - on what learners can define e.g. importance etc. Teachers and instructors have a broader view of the profession, so we have them!

Particular comments

  1. The issues discussed in this paper attempt to integrate the computerized learning objectives, and create specifications that allow multiple instructors to work together to develop valuable and reconfigurable learning contents. Explanations???

  1. standardizing educational materials becomes increasingly important? What it means standardized you talk about (flexible)? The claims are not consistent - additional explanations are needed!

  1. The most important functionality for an E-learning 2.0 platform is the process which adaptively provides a versatile course portfolio scheme for different learners from different perspectives. Additional explanations needed!

  1. But, with the help of our approach with E-learning 2.0, S may proactively choose some of the topics in marketing and some of the topics in database management to customize his own course portfolio (may be called Database Marketing) by the course portfolio mechanism as illustrated in Figure 4. My fundamental dilemma is, How does a learning person know what they needs, what knowledges are primary?

  1. Correlations between Figure 4 and Figures 7 and 8 (role of instructors is not clear enought?)

Illustrative Example

  1. satisfying each criterion on the second level - which criterion - additional explanation needed!

  1. weighting values - who define and how?

  1. Automatic Course Portfolio Analysis - additional explanation needed!

Reviewer 2 Report

The presented article proposes a flexible framework for students to personalize their learning in a digital environment. They propose a hierarchical analysis framework (AHP) to build adaptive portfolios and place great importance on the active role of students in this process. The research is very significant and relevant. In relation to the criteria requested by the journal for the publication of the article, I must state the following aspects to take into account:
1º In Pedagogy there are many systematic reviews on Personal Learning Projects (PLE), it would be important for the authors to refer to the existing literature on the subject. Some authors who have worked on this issue.

-          Attwell, G. (2007). Personal Learning Environments-the future of eLearning. Elearning papers, 2(1), 1-8.-          Sangrà, A., Raffaghelli, J.E. and Veletsianos, G. (2019), “Lifelong learning Ecologies: Linking formal and informal contexts of learning in the digital era”. Br J Educ Technol, 50: 1615-1618. doi:10.1111/bjet.12828
-          Siemens, G. (2007). Connectivism: Creating a learning ecology in distributed environments. In Th. Hug (Ed.), Didactics of microlearning: Concepts, discourses, and examples (pp. 53-68). Múnster:  Waxmann Verlag.
-          Yurkofsky, M.M., Blum-Smith, S., & Brennan, K. (2019). Expanding outcomes: Exploring varied conceptions of teacher learning in an online professional development experience. Teaching and Teacher Education, 82, 1-13. https://doi.org/10.1016/j.tate.2019.03.002
2º The research design is a framework where an analysis of the courses and the contents is carried out through hierarchical analysis (AHP). Is this the research design? How are the results validated?

3rd If the general objective is to seek personalization of the students' learning processes, where do we find their opinions or choices?

Round 2

Reviewer 1 Report

The paper is improved, could be better but from my point of view they reach minimal standards.